# Impact of Functional Training on Functional Movement and Athletic Performance in College Dragon Boat Athletes

**DOI:** 10.3390/ijerph20053897

**Published:** 2023-02-22

**Authors:** Chichong Wu, Manwai Cheong, Yan Wang, Xiuqiang Wang, Qingwen Zhang, Minghui Li, Siman Lei

**Affiliations:** 1Choi Kai Yau College, University of Macau, Macau SAR, China; 2School of Physical Education, Shanghai University of Sport, Shanghai 200433, China; 3Faculty of Education, University of Macau, Macau SAR, China; 4Exercise, Health and Technology Centre, Department of Physical Education, Shanghai Jiao Tong University, Shanghai 200240, China; 5Exercise Translational Medicine Centre, Shanghai Centre for Systems Biomedicine, Shanghai Jiao Tong University, Shanghai 200240, China; 6Department of Sports Science and Physical Education, The Chinese University of Hong Kong, Hong Kong SAR, China

**Keywords:** functional training, athletic performance, dragon boat, athletes

## Abstract

Functional training has become a popular training method in different sports, yet limited studies have focused on paddle sports. The purpose of this study was to evaluate the effects of functional training on functional movement and athletic performance in college dragon boat athletes. A total of 42 male athletes were divided into 2 groups: a functional training (FT) group (n = 21, 21 ± 1.47 years) and a regular training (RT) group (n = 21, 22 ± 1.50 years). The FT group participated in an 8-week (16-session) functional-training program, while the RT group trained with strength-training sessions. Functional movement screen (FMS), Y-balance test (YBT) and athletic performance evaluations were conducted before and after the intervention. Repeated measure ANOVA and *t*-test evaluations were employed to examine differences for both groups. The FT group was significantly improved in FMS scores (F = 0.191, *p* < 0.001) and YBT scores (F = 2.59, *p* = 0.027), and it also showed significantly improved muscular fitness (pull-ups: F = 0.127, *p* < 0.001; push-ups: F = 1.43, *p* < 0.001) and rowing speed (F = 4.37, *p* = 0.004). It is recommended to include functional training as a part of training and routine exercise, as it appears to be an effective way of improving FMS and athletic performance in paddle sports.

## 1. Introduction

In Asia-Pacific, dragon boating has progressed from a leisure festival event to a social and competitive sport [1]. It has been acknowledged as an international competitive team-based water sport [2]. Winning a high-level tournament in dragon boat racing requires proper paddling technique and posture. A boat that is moved by paddling force is also affected by stroke movement [3]. In addition, because training and competition are physically demanding, incorrect positions and training loads are more likely to increase the risk of injury [4].

Even though there are many different types of training, including aerobic, concurrent, and strength training, functional training is thought to be able to enhance athletic performance by enhancing core strength, balance, coordination, and movement patterns [5,6]. Early functional training was used in physical therapy and focused on the rehabilitation of the elderly or those with medical conditions [7,8,9,10]. Over the years, functional training has developed rapidly, with a trend shifting to athletics areas that focus on performance and injury reduction [11,12]. Functional training can improve power transmission and stabilize body movement during paddle actions [13]. Compared to rowing and kayaking, dragon boat racing in flat water and the unilateral movement of transferring power are uneven, which can affect performance and cause various sports injuries. Therefore, it is vital to enhance functional movement and maintain stability for improving paddling technique (catch, drive, and recovery) [3]. Unlike traditional strength training, which is theoretically focused on individual muscles or muscle groups, functional training is not focused on specific muscles but rather on movement patterns, stability, and strength [14]. Functional training also emphasizes body movement as a whole that engages with core strength, multidimensional, and multiplane movement. This training program also follows a progressive principle, from basic to complex, from normal function to specific function [7,15]. Furthermore, it is based on functional movement screen (FMS) evaluations that examine movement qualities and weaknesses [5]. In addition, athletic performance is evaluated as a reference for customizing functional exercises [6].

Several studies have applied functional training to enhance performance across different sports [16,17,18,19,20,21,22]. A systematic review concluded that functional training improved athletic performance and fitness [12]. Moreover, athletic fitness is considered a critical and fundamental competence for evaluating competitive ability. Studies have found that athletic performance and functional training are positively correlated [18,19,23]. Athletic performance can also be improved through athletic fitness and sport performance [18,24,25,26,27]. To achieve training efficiency, purposeful and systematic training determined under a series of functional and fitness profiles can build a successful team [6]. Dragon boat races range from 200 m to 2000 m, and the two most popular race distances in these events are 200 m and 500 m at the international level. Developing successful training plans for different race distances should be based on the aerobic and anaerobic energy requirements of each distance [28]. Thus, the training program in this study aims to increase the maximal energy capacities in 200 m and 500 m races.

Numerous studies on the effect of functional training have shown improved functional movement and sports performance in physical education students, soccer players, and a variety of athletes [16,19,21,29,30]. Given the significance of stability and mobility in the physical performance of rowing sports and the paucity of data on the influence of functional training on college paddlers, this study hypothesizes that an 8-week functional training program can improve the FMS and athletic performance of college dragon boat athletes. We also expect to provide recommendations and directions for coaches to conduct well-rounded functional programs.

## 2. Materials and Methods

### 2.1. Study Design and Participants

This study was a controlled, nonrandomized trial designed to assess whether functional training could improve the levels of FMS and athletic performance in dragon boat athletes.

A total of 44 male dragon boat athletes were recruited from two universities in Macau SAR, China. Due to the COVID-19 epidemic, all subjects did not engage in trainings for more than three months and did not train using functional training before the experiment. Two participants from both group were eliminated in the process of the study (Figure 1). All athletes competed in 200 and 500 m races in the university category of the International Dragon Boat Competition. Participants were divided into two groups: the functional training (FT, n = 21, 21 ± 1.47 years) group and the regular training (RT, n = 21, 22 ± 1.50 years) group. A questionnaire was used to analyze the demographic background of participants. Voluntary participants were invited to participate and provided written consent after learning about the process and risks of the study.

The inclusion criteria for the study were as follows: (1) college dragon boat athlete; (2) male; and (3) no past sustained musculoskeletal surgery of any kind. Subjects who met the following criteria were excluded: physically restricted or medically restricted (Figure 1).

This research was approved by the Ethics Committee for research involving human participants of the University of Macau (SSHRE21-APP047-FED) and was conducted according to the Declaration of Helsinki.

### 2.2. Functional-Training Program

A trial period was arranged for the FT and RT groups to present the training method, for demonstration, and for movement familiarization. Both groups continued twice a week on nonconsecutive days for eight weeks (16 sessions). Two trained professionals executed the training. As part of the functional-training program, exercises were included that complemented the multi-plane and multi-joint movements that are used in dragon boat strokes. Additionally, the functional-training program focused on functional movement, stability, mobility, and core strength (Table 1). Consequently, these exercises were designed based on the literature and previous functional training, as well as taking into account the results of the FMS test and other relevant fitness measurements [6,7,19,25,26,30,31,32], and were set based on functional-training principles [6,7,32]: (1) from primary to functional; (2) from single-joint to multiple-plane stages; and (3) from training multiple muscles to improving whole-body function.

The FT group (n = 21) participated in land-based functional training, while the RT group (n = 21) performed a land-based warm-up and regular strength exercises (e.g., free weight training) before practicing in the water. The intervention consisted of 3 phases (weeks 1–2, weeks 3–4, and weeks 5–8), with 60 min per session (2 sessions per week) divided into an initial 10 min of dynamic stretching, 40 min of functional training, and 10 min of recovery.

Every functional-training session consisted of ten exercises executed in three sets of 16–20 repetitions with a rest period of 50–60 s between sets. The number of repetitions was evaluated based on the training level and intensity. Functional movement capacity was developed using body weight, elastic resistance bands, medicine balls, and BOSU balls to maintain training intensity.

The intensity of the training program was monitored by heart rate and the rated perceived exertion (RPE) scale [33,34]. A Polar OH1 heart rate monitor was utilized to determine the instance training loads [35] and intensity levels [36]. The average heart rates during the training were recorded at 142 ± 6.2 bpm (Phase 1), 147 ± 6.8 bpm (Phase 2), and 149 ± 6.4 bpm (Phase 3). The rated perceived exertion scale was used to evaluate subjective perspective about the training intensity on a 10-point scale from 0 to 10 [34]. The average RPE values were recorded at 6.9 ± 1.3 (Phase 1), 7.5 ± 1.6 (Phase 2), and 7.8 ± 1.7 (Phase 3). The program intended to maintain intensity between moderate and hard levels.

### 2.3. Evaluation

The evaluation was conducted by trained surveyors. An anthropometric analysis and body composition were measured using bioelectrical impedance (Inbody 270 model) [37]. Functional moments were measured using FMS, which evaluated individual movement function, deficiency, stability, and asymmetry. The FMS evaluation consisted of 7 movement pattern tests, including functional movement (deep squat, hurdle step, and in-line lunge), fundamental mobility (shoulder mobility and active straight-leg raise), fundamental core strength (trunk push-up), and fundamental core stability (rotary stability) [13]. The scoring mechanism used a standard 0–3 ordinal scale to grade the quality of a movement in compliance with a standard [38] and could receive a total score of between 0 and 21 points. The Y-balance test (YBT) measured dynamic balance by calculating the absolute reach distance in three directions, as well as the comprehensive movement coordination of all domains of movements, including strength, core stability, and the motion range of a subject [39,40].

Athletic performance was measured by the following: (1) muscle strength and endurance (handgrip, push-ups, and pull-ups), with a handgrip test measuring the maximum grip for each hand [41], push-ups referring to the maximum number of push-ups completed within one minute [42], and pull-ups referring to the maximum number of pull-ups completed at one time [43]; (2) power, as evaluated by a standing long jump with the total jumping distance [44]; (3) flexibility, as tested by a sit-and-reach exercise that reached forward to the maximum distance [45]; (4) agility, as measured by a shuttle run that was repeated four times over a distance of 10 m [46]; (5) speed, as measured by a 30 m sprint that reached maximum speed within the distance [47]; and (6) rowing speed, as evaluated by rowing 200 m on an indoor rowing ergometer [48].

### 2.4. Statistical Analysis

Statistical analyses were carried out using excel version 2019 and SPSS version 27.0. The collected data were presented ass mean ± standard deviation (SD), and a significance level of *p* < 0.05 was established A *t*-test was employed to assess differences in baseline characteristics between the FT and RT groups. Moreover, *t*-test and two-way repeated ANOVA evaluations were conducted to determine the differences in the vitals measured within and between the pre- and post- intervention protocols. Cohen’s effect sizes were calculated for the differences between the FT and RT groups; the effect sizes (ESs) were interpreted as “small” (0.2–0.49), “medium” (0.5–0.79), and “large” (>0.8) [49]. In this study, all coefficients were accepted as statistically significant at 95% (*p* < 0.05). The sample size analysis was calculated utilizing G*power calculation [50]. This study was performed with a moderate effect size of 0.5, an alpha error of 0.05, and a desired power (1-ß error) of 0.95.

## 3. Results

A total of 44 individuals were recruited for this study, two individuals were eliminated. The remaining 42 individuals were divided into two groups. No significant difference between the two groups were identified in the characteristics and anthropometric factors prior to the experiment, including age, height, weight, BMI, body fat, and muscle mass (Table 2). Neither group reported side effects during or after the experiment.

FMS, YBT, and athletic performance parameters were analyzed with *t*-test and two-way repeated measure ANOVA evaluations. The FT group significantly increased (F = 0.191, *p* < 0.001), while the RT group slightly raised in FMS scores. The FT group found significant differences in two subsections of FMS: active leg raises (F = 2.84, *p* = 0.020) and trunk push-ups (F = 0.144, *p =* 0.021). After the intervention, functional movement (deep squat, hurdle step, and in-line lunge), fundamental mobility (shoulder mobility), and fundamental core stability (rotary stability) in the FT group increased (Table 3). The effect size for the FMS scores was 0.78, while that for active leg raises was 0.63 and that for trunk push-ups was 0.52.

The YBT was calculated based on the maximum reach in the anterior, posteromedial, and posterolateral directions. YBT scores were significantly improved (F = 2.59, *p =* 0.027), and the left stance limb of the YBT was significantly increased (F = 1.82, *p =* 0.008) in the FT group.

In terms of athletic performance, the selected parameters are presented in Table 4. Those of the FT group were significantly improved in push-ups (F = 0.127, *p* < 0.01), pull-ups (F = 1.43, *p* < 0.01), and rowing speed (F = 4.37, *p =* 0.004). Agility (*p* < 0.05), speed (*p =* 0.054), and power (*p =* 0.009) were found to be significantly different between the FT and RT groups (Table 4). Medium-to-large effect sizes were found for pull-ups, push-ups, and 200 m rowing, and the Eta squared values were 0.52, 0.75, and 0.85, respectively.

## 4. Discussion

The primary findings of this study indicate that 8 weeks of functional training improved the functional movement abilities and athletic performances of college athletes.

The effect sizes for FMS, the YBT, push-ups, pull-ups, and rowing speed ranged from medium to large.

The findings supported the hypothesis that functional training significantly improves overall functional movement, fundamental mobility, core strength, muscular fitness (pull-ups and push-ups), and rowing speed. Even though the RT group improved in FMS, the YBT, muscular fitness (handgrip), agility, speed, flexibility, and power, the improvements were not significant. Significant differences between the groups were found in the parameters of agility, speed, and power. Although there were various intervention programs, this 8-week intervention enhanced functional movement ability and athletic performance similarly to previous studies [19,31,51]. Functional training has been proved to be effective for youth, college, and semi-professional athletes involved in a variety of sports [16,17,20,21,22,25,26,30]. To the best of our knowledge, limited studies have examined the effectiveness of functional training among college dragon boat athletes and paddle sport athletes, and this study can serve as a primary resource for paddle sport athletes and coaches.

Successful racing teams have better results for force development during water entry, drive force, force reduction during paddle exit, stroke length, and paddling stroke [3]. These movements are associated with the performance of the kinetic chain, which requests a great number of dorsal, abdominal, and limb muscles. The functional training in this study improved functional movement and fundamental mobility by stabilizing muscles and increasing joint range of motion, which may improve stroke technique ability. A cross-sectional study on FMS and the performance of paddle sport athletes [52] found that racing time was correlated with overall FMS scores. It also mentioned that functional movements were associated with race performance. The findings of this study could provide support for the effects of functional training on the functional movement, strength development, and sport performance of paddle sport athletes. Functional training involves exercises that engage in sport-specific performances, such as pushing, pulling, and rotational movement, that are specially designed for dragon boating. Functional training is intended to help an individual grow stronger and more efficient, as well as to improve coordination, balance, and stability.

### 4.1. Functional Movement Ability

The FT and RT groups both showed improved FMS scores over 8 weeks, with a significantly improvement noted in the FMS scores of the FT group from 14.81 ± 1.30 to 16.86 ± 1.28 (*p* < 0.001). Thus, it could be claimed that the FMS score baseline of scores over 14 is more likely to prevent injuries [11], and may benefit athletic fitness and performance development. Functional training could significantly enhance functional movement ability, which is consistent with previous findings [12,19,20,21,22,25,26,30,31].

The intervention positively affected overall functional movement ability; it also improved the areas of fundamental mobility and fundamental core strength. Active leg raises and trunk push-ups were identified to have significant differences within the FT group. Active leg raises increased from 1.71 ± 0.46 to 2.29 ± 0.56 (*p =* 0.021); the better the performance for active leg raises, the better the reflection of the flexibility of the hamstring, the hip mobility, and the lower core stability [38]. The stability of the hip and lower core are important for seating position in dragon boating [3]. Trunk push-ups increased from 2.14 ± 0.48 to 2.71 ± 0.46 (*p =* 0.02) in the FT group after the intervention. It can said that trunk push-ups reflect trunk stability, which is an essential ability for transferring force between the upper and lower extremities [38]. Trunk stability can affect trunk rotation, which can affect the stroke and paddling techniques of dragon boating [3].

The YBT assessed neuromuscular control and dynamic balance [53]. It was found that performing the YBT over longer distances could stabilize the body during different movement [40,54]. The YBT score was increased from 92.2 ± 5.60 cm to 96.6 ± 4.88 cm (*p =* 0.027) in the FT group, which indicates that functional training could improve dynamic balance. The left stance limb of the YBT was also significantly improved (*p =* 0.008) after the intervention. A greater performance on the YBT could lead to improve balance control and more even transfer of force through the lower extremities [55].

### 4.2. Athletic Performance

Functional training is used to prevent injury risk and improve movement patterns and is also used to improve athletic performance [24]. This study showed that functional training significantly improved muscular fitness and rowing speed in the FT group. The findings are consistent with recent studies showing that functional training significantly improves different athletic performance variables [16,17,18,19,25,26,30,31]. Although there were improvements in agility, speed, and power in the FT group, these parameters were significantly different between the FT and RT groups.

Muscular fitness is fundamental for paddle sports, as repetitive movement requires strength and endurance. In a case of dragon boating, racing distance varies from 200 to 2000 m, requiring short-to-long-term strength and endurance. The intervention program of this study highly affected muscular fitness. The number of pull-ups increased from 7.29 ± 4.48 to 10.00 ± 3.99 (*p* < 0.001) in the FT group, consistent with functional trainings that have improved muscular strength [17,24,46]. The push-ups increased from 37.67 ± 10.76 to 47.86 ± 10.46 (*p* < 0.001), which is consistent with a study of functional training that improved muscular endurance [47]. Dragon boating involves a variety of muscles. Stronger muscular strength and endurance are necessary to stabilize and sustain the stroke movement.

Dragon boat racing is a competitive team sport where each athlete performs an effective stroke to propel a boat forward on water. The results of the 200 m timed trial on the rowing ergometer significantly improved in the FT group, changing from 46.52 ± 1.50 to 44.52 ± 1.91 s (*p =* 0.04). The results are consistent with functional training that could increase rowing speed after intervention [17]. Another FMS and paddle sport study emphasized that the relationship between FMS score and racing speed was significantly related [52]. There is evidence to suggest that functional training can enhance rowing speed and may serve as a key indicator for other paddle sports.

Several limitations need to be acknowledged. (1) This was a nonrandomized trial in which participants were divided into two groups. Although the baseline characteristics of the athletes were not significantly different, a random control trial could be applied in future studies. (2) This study focused on male college dragon boat athletes, but it has the potential to extend functional training to other groups (e.g., female or elite). (3) Although the repetitions had ranges for adaptation to individuals, it is also suggested to design individual profiles for training in further research.

## 5. Conclusions

This study demonstrated the effects of a functional intervention on improving FMS, YBT, and athletic performance. This intervention could have practical benefits for various paddle sports, including cost-effective implementation and less restriction on training space and equipment. It is valuable to understand that functional training could not only prevent injury risk, but also could enhance the ability of athletic performance. The outcomes are likely change with the selected exercises, intervention duration, and training program used. Additionally, measurable enhancements occurred within 8 weeks of intervention. This may encourage athletes or sports teams to train with a functional-training program.

## Figures and Tables

**Figure 1 ijerph-20-03897-f001:**
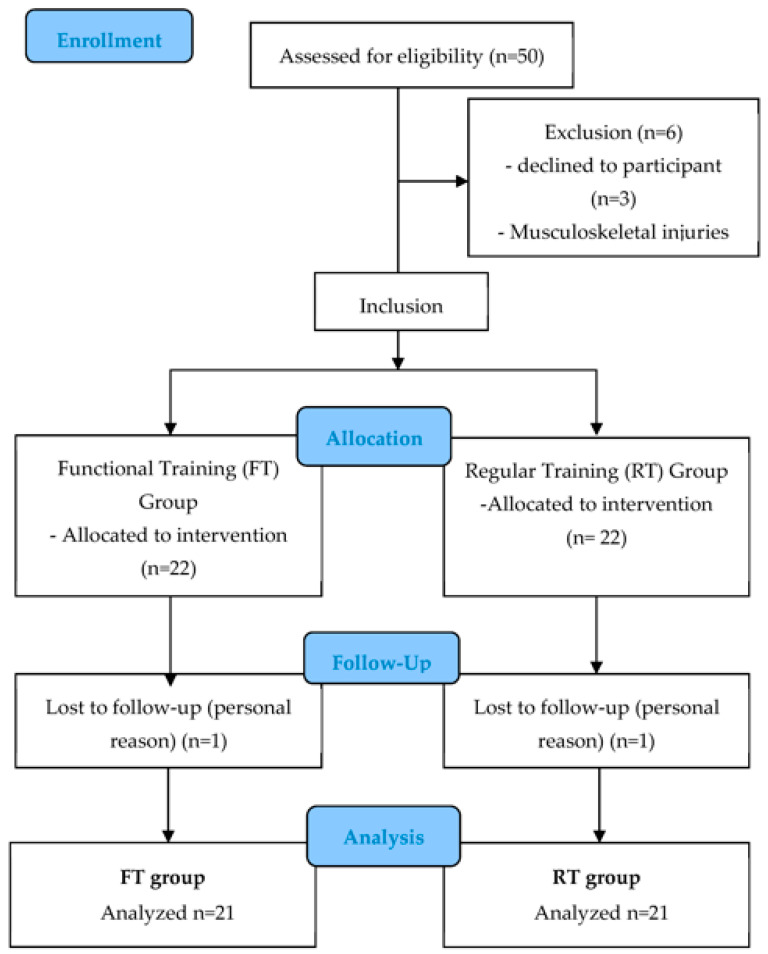
Flow chart of the experiment design.

**Table 1 ijerph-20-03897-t001:** Functional-training program.

Category	Phase 1 (Weeks 1–2)	Phase 2 (Weeks 3–4)	Phase 3 (Weeks 5–8)
	Static stretch, dynamic stretch		
Upper	Explosive push-up	Single-leg push-up	Single-leg push-up (MB)
	Low kneeing chop	Standing chop	Multidirection chop
	Throw (MB)	Side throw (MB)	Multidirection throw (MB)
	Rowing (RB)	Single-arm rowing (RB)	Side bridge rowing (RB)
Core	Plank	Single-leg plank	Single-leg plank (BB)
	Mountain climber	Cross-body mountain climber	Mountain climber (BB)
	Russian twist	Russian twist (MB)	Russian twist (BB and MB)
	Dead bug	Dead bug (BB)	Dead bug with pause (BB)
Lower	Squat (RB)	Squat (BB)	Single-leg squat
	Lunge	Single-leg lunge	Lunge jump
	Recovery		

Note: RB, resistance band; MB, 4 Kg medicine ball; BB, BOSU ball.

**Table 2 ijerph-20-03897-t002:** Athlete characteristics (n = 42).

	FT Group (n = 21)		RT Group (n = 21)	
	Pre	Post	Pre	Post
Age	21 ± 1.47		22 ± 1.50	
Height	174.7 ± 5.32		173.9 ± 6.24	
Weight	71.7 ± 6.14	71.9 ± 5.52	70.6 ± 8.76	70.5 ± 8.82
Body Mass Index	23.5 ± 2.26	23.7 ± 2.12	23.3 ± 2.27	23.3 ± 2.25
Skeletal Muscle Mass	33.4 ± 3.31	33.6 ± 3.29	33.3 ± 3.40	33.2 ± 3.56
Percent Body Fat	17.5 ± 4.60	17.6 ± 4.20	16.3 ± 6.36	16.6 ± 6.56

**Table 3 ijerph-20-03897-t003:** YBT and FMS data from FT and RT groups.

Parameters	Test	Group				*p* (Between Groups)	*η* ^2^
YBT		FT (n = 21)	*p*	RT (n = 21)	*p*		
YBT scores	Pre	92.2 ± 5.60	0.027 *	96 ± 4.09	0.356	0.109	0.51
	Post	96.6 ± 4.88		97.3 ± 4.93			
YBT-Left	Pre	91.92 ± 5.78	0.008 *	95.99 ± 4.58	0.805	0.185	0.48
	Post	96.55 ± 5.64		96.35 ± 4.98			
YBT-Right	Pre	92.46 ± 5.97	0.196	95.95 ± 4.14	0.130	0.07	0.29
	Post	96.69 ± 4.42		98.19 ± 5.21			
Functional Movement							
FMS scores	Pre	14.81 ± 1.30	<0.001 **	15.48 ± 1.37	0.260	0.665	0.78
	Post	16.86 ± 1.28		15.90 ± 1.04			
Deep Squat	Pre	2.33 ± 0.58	0.776	2.19 ± 0.51	0.358	0.426	0.28
	Post	2.38 ± 0.50		2.33 ± 0.48			
Hurdle Step	Pre	2.43 ± 0.50	0.367	2.38 ± 0.59	0.780	0.475	0.22
	Post	2.57 ± 0.50		2.43 ± 0.51			
In-Line Lunge	Pre	2.19 ± 0.60	0.270	2.19 ± 0.60	0.771	0.585	0.32
	Post	2.38 ± 0.50		2.44 ± 0.44			
Shoulder Mobility	Pre	2.14 ± 0.73	0.147	2.24 ± 0.59	0.796	0.851	0.25
	Post	2.43 ± 0.51		2.29 ± 0.64			
Active Leg Raise	Pre	1.71 ± 0.46	0.021 *	2.24 ± 0.70	0.814	0.101	0.63
	Post	2.29 ± 0.56		2.19 ± 0.60			
Trunk Push-up	Pre	2.14 ± 0.48	0.020 *	2.29 ± 0.46	0.213	0.706	0.52
	Post	2.71 ± 0.46		2.48 ± 0.51			
Rotary Stability	Pre	1.86 ± 0.36	0.088	1.95 ± 0.38	1.000	0.813	0.23
	Post	2.10 ± 0.30		1.95 ± 0.50			

Note: FT, functional training; RT, regular training. Mean ± standard error; level of significance: * *p* < 0.05 and ** *p* < 0.001.

**Table 4 ijerph-20-03897-t004:** Athletic performance data from FT and RT groups.

Parameters	Test	Group				*p* (Between Groups)	*η* ^2^
Muscular fitness		FT (n = 21)	*p*	RT (n = 21)	*p*		
Hand grip (KG)	Pre	42.31 ± 8.31	0.921	39.99 ± 8.23	0.459	0.486	0.32
	Post	42.54 ± 6.70		41.85 ± 7.89			
Pull-up	Pre	7.29 ± 4.48	<0.001 **	8.29 ± 4.54	0.884	0.724	0.52
	Post	10.00 ± 3.99		8.10 ± 3.87			
Push-up	Pre	37.67 ± 10.76	<0.001 **	39.95 ± 11.02	0.606	0.238	0.75
	Post	47.86 ± 10.46		38.38 ± 8.39			
Agility							
4 × 10 m shuttle run	Pre	11.53 ± 1.45	0.06	10.88 ± 0.69	0.105	0.05 *	0.23
	Post	10.53 ± 0.67		10.53 ± 0.67			
Speed							
30 m sprint (S)	Pre	5.10 ± 0.45	0.781	4.87 ± 0.40	0.796	0.054 *	0.33
	Post	5.05 ± 0.48		4.84 ± 0.40			
Flexibility							
Sit and reach (cm)	Pre	12.15 ± 4.09	0.122	13.27 ± 7.55	0.570	0.621	0.43
	Post	14.31 ± 4.77		14.70 ± 8.57			
Power							
Standing long jump	Pre	198.48 ± 23.39	0.121	223.67 ± 27.24	0.821	0.009 *	0.35
	Post	215.52 ± 27.12		225.52 ± 25.56			
Rowing speed							
200 m rowing	Pre	46.52 ± 1.50	0.004 *	46.26 ± 2.28	0.651	0.043 *	0.85
	Post	44.52 ± 1.91		46.52 ± 1.33			

Note: FT, functional training; RT, regular training. Mean ± standard error; level of significance: * *p* < 0.05 and ** *p* < 0.001.

## Data Availability

The data used and analyzed during this study are available upon reasonable request from the corresponding author.

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
