# Peer review of "Impact of Functional Training on Functional Movement and Athletic Performance in College Dragon Boat Athletes"

_ijerph, 2023, doi:10.3390/ijerph20053897_

Round 1
Reviewer 1 Report
Abstract
Lines 20-21 - I imagine that the numbers in parentheses are the age, please identify (ex: 21, years old).
The results about ANOVA is not reportes at abstract and it should be.
Introduction
In this section, the focus should be on functional training. Try to follow a logic: start by describing the modality, functional training and finally what would be the importance of functional training in the chosen modality. As you did not evaluate imbalance, I suggest not talking about it in the introduction.
Materials and Methods - Study design and participants:
Lines 84-86 - This information (No significant difference between two 84 groups were identified in characteristics and anthropometric factors prior to the experi- 85 ment, including age, height, weight, BMI, body fat, and muscle mass) is a result.
Lines 86-88 - This information (The sample size 86 analysis was calculated by utilizing G*power calculation[30]. This study was performed 87 with a moderate effect size=0.5, an alpha-error=0.05, and the desired power (1-ß er- 88 ror)=0.95) is statistical aspects.
Did the participants already perform some type of functional training? or strength training? How was the training routine of these athletes before the intervention? in the intervention, did they stop carrying out the activities they used to do and only carry out the training proposed by the study? this all must be described.
Results
Lines 217-218 - This is statistical aspects (FMS, YBT, and athletic performance parameters were analyzed by two-way repeated measure ANOVA).
You should report the p' values of RT training too.
Discussion
It was not very clear how different the FT was from the RT, so the discussion should also be directed towards comparing the results of the two types of training.
Author Response
Dear Respected Editor and reviewers,
Thank you so much for providing us so many constructive suggestions and an opportunity to revise our manuscript. Responses to reviewers are provided in a point-by-point manner. All of the additions and changes have been highlighted in red in the re-submission file.
Reviewer-1
Abstract
Lines 20-21 - I imagine that the numbers in parentheses are the age, please identify (ex: 21, years old).
Response: Thanks for your kind suggestion as it is important to present the figure with standard format. We have added relevant details in Line 20-21.
- The results about ANOVA is not reportes at abstract and it should be.
Response: Thanks for your kind suggestion as it is important to present the figure with standard format. We have added relevant details in Line 25-27.
Introduction
- In this section, the focus should be on functional training. Try to follow a logic: start by describing the modality, functional training and finally what would be the importance of functional training in
the chosen modality.
Response: Thanks for your kind suggestion as it is important to describe it with a logic. We have amended relevant details in Line 40 -43.
4. As you did not evaluate imbalance, I suggest not talking about it in the introduction
Response: Thanks for your kind suggestion as we did not evaluate imbalance. We did not mentioned imbalance in the introduction.
Materials and Methods - Study design and participants:
5. This information (No significant difference between two 84 groups were identified in characteristics and anthropometric factors prior to the experi- 85 ment, including age, height, weight, BMI, body fat, and muscle mass) is a result.
Response: Thanks for your indication. We have changed this part to Result in Line 219-221.
- This information (The sample size 86 analysis was calculated by utilizing G*power calculation[30]. This study was performed 87 with a moderate effect size=0.5, an alphaerror=0.05, and the desired power (1-ß er- 88 ror)=0.95) is statistical aspects.
Response: Thanks for your indication. We have changed this part to Statistical analysis in Line 214-216.
7. Did the participants already perform some type of functional training? or strength training? How was the training routine of these athletes before the intervention? in the intervention, did they stop carrying out the activities they used to do and only carry out the training proposed by the study? this all must be described
Response: Thanks for pointing it out. We have added the relevant content about training before the experiment. The addition can be seen in Line 88 -89.
Results
8. Lines 217-218 - This is statistical aspects (FMS, YBT, and athletic performance parameters were analyzed by two-way repeated measure ANOVA)
Response: Thanks for pointing it out. We have mentioned the relevant content about the statistical aspects in Line 208-211
- You should report the p' values of RT training too.
Response: Thanks for your suggestion. p value was added to Table 3 and Table 4. Please refer to Line 238 and 249
Discussion
- It was not very clear how different the FT was from the RT, so the discussion should also be directed towards comparing the results of the two types of training.
Response: Thanks for your kind suggestion. We should emphasis the difference between FT and RT group. The addition can be seen in Line 260-263
Reviewer 2 Report
I found the paper quite interesting. The strength of the article is the well-described methodology and intervention. The authors used the correct statistical methods and exhaustively described them in the paper.
The results are clearly described. The discussion section could be extended with a deeper explanation of the impact of the methods and exercises used on the performance and functional usefulness of Dragon Boat Athletes. I wonder about the results of the control group. The authors mentioned that the control group performed standard strength exercises in place of the functional intervention. On the other hand, the graphs show a decrease in the strength values in the RT group.
Below I have included some editorial notes that I believe will add value to the work.
On lines 35 - before [2]; 87 - before [30]; 190 - before [41] add a space
Line 145 - remove dot before ". (16 sessions)"
In Table 1 - Medicine ball throw, the throw was made from behind the head with both hands? and what was the mass of the ball? - add this information
Line 170-175 - add standard deviation to mean HR and mean RPE
183-184 – the FMS was assessed by one person? was it a trained person? was the subjective assessment carried out collegially?
"p" in p-value should be italitics
add 0 in p-value - p < 0.001
Author Response
Dear Respected Editor and reviewers,
Thank you so much for providing us so many constructive suggestions and an opportunity to revise our manuscript. Responses to reviewers are provided in a point-by-point manner. All of the additions and changes have been highlighted in red in the re-submission file.
Reviewer-2
- The discussion section could be extended with a deeper explanation of the impact of the methods and exercises used on the performance and functional usefulness of Dragon Boat Athletes.
Response: Thanks for your kind suggestion. It is important to extend a deeper explanation in the discussion. We have added relevant details in 271 – 285.
- I wonder about the results of the control group. The authors mentioned that the control group performed standard strength exercises in place of the functional intervention. On the other hand, the graphs show a decrease in the strength values in the RT group.
Response: Thank you for pointing it out. When we designed the training program for both group, we try to avoid the training program that are similar to the evaluated parameter, therefore the strength program was conducted with free weight exercise. However, the change of RT group was mentioned in discussion in Line 260 to 262.
- On lines 35 - before [2]; 87 - before [30]; 190 - before [41] add a space Line 145 - remove dot before ".(16 sessions)"a
Response: Thank you for pointing it out. It is important to standardize the format. We have amended the relevant information.
- In Table 1 - Medicine ball throw, the throw was made frombehind the head with both hands? and what was the mass of the ball?
Response: Thank you for suggestion. We have amended the relevant information in Line 169
- - add this information Line 170-175 - add standard deviation to mean HR and mean RPE 183-184
Response: Thank you for indication. We have amended the relevant information in Line 176-177
- – the FMS was assessed by one person? was it a trained person? was the subjective assessment carried out collegially?
Response: Thank you for indication. We have amended the relevant information in Line 182.
- "p" in p-value should be italitics
Response: Thank you for indication. We have amended the relevant information throughout the paper
- add 0 in p-value - p < 0.00
Response: Thank you for indication. We have amended the relevant information in Line 240 and Line 253
Reviewer 3 Report
The purpose of this study is to evaluate the effect of functional training on the functional exercise and performance of college dragon boat athletes. From the perspective of topic selection, this is a relatively small topic, so the author needs to describe the differences and links between dragon boat athletes and other sports. Secondly, the sample size in this study is only 42 male athletes. As a research sample, the number may be relatively small. Secondly, as a research sample, I suggest that the female sample should be supplemented, so that there can be a comparison, which is more meaningful for dragon boat training. In terms of research conclusions, this study seems to only reveal the significance of data, and it is suggested that the author can further explore the mechanism behind the data. At the same time, it is suggested that the author should also make a comparison between good areas or school dragon boat teams, because the methods and results adopted by athletes at different levels are different. Good luck.
Author Response
Dear Respected Editor and reviewers,
Thank you so much for providing us so many constructive suggestions and an opportunity to revise our manuscript. Responses to reviewers are provided in a point-by-point manner. All of the additions and changes have been highlighted in red in the re-submission file.
- The purpose of this study is to evaluate the effect of functionaltraining on the functional exercise and performance of collegedragon boat athletes. From the perspective of topic selection,this is a relatively small topic, so the author needs to describethe differences and links between dragon boat athletes and other Secondly, the sample size in this study is only 42 male athletes. As a research sample, the number may be relativelysmall.
Response: Thank you for raising these important points. . The study originally recruited 50 participants, and excluded 6 participants due to serval reasons, eventually 44 participants joined the experiment. The sample size analysis was calculated by utilizing G*power calculation. It was performed with a moderate effect size=0.5, an alpha-error=0.05, and the desired power (1-ß error)=0.95.
- Secondly, as a research sample, I suggest that the femalesample should be supplemented, so that there can be acomparison, which is more meaningful for dragon boat training.
Response: Thank you for your comments. That would be more meaningful to include female athletes as a part of sampling. The study focused on the college dragon boat athletes, but Macau does not have a college dragon boat category for female athletes. We have also mentioned it in the limitation.
- In terms of research conclusions, this study seems to only revealthe significance of data, and it is suggested that the author canfurther explore the mechanism behind the data. At the sametime, it is suggested that the author should also make acomparison between good areas or school dragon boat teams,because the methods and results adopted by athletes at differentlevels are different.
Response: Thank you for your suggestion. I appreciate your advice and I will consider it in my future research. I plan to compare athletes of varying levels to assess the impact of this data, as well as analyze the mechanism behind it. This could provide more meaningful insight into the significance of the data observed.
Round 2
Reviewer 1 Report
-
Reviewer 3 Report
The author revised the article very well. The article has met the requirements of publication. It is recommended to accept. good luck.